# Anticancer Activity of Chalcones and Its Derivatives: Review and In Silico Studies

**DOI:** 10.3390/molecules28104009

**Published:** 2023-05-10

**Authors:** Fernando Ferreira Leite, Natália Ferreira de Sousa, Bruno Hanrry Melo de Oliveira, Gabrielly Diniz Duarte, Maria Denise Leite Ferreira, Marcus Tullius Scotti, José Maria Barbosa Filho, Luís Cezar Rodrigues, Ricardo Olímpio de Moura, Francisco Jaime Bezerra Mendonça-Junior, Luciana Scotti

**Affiliations:** 1Post-Graduate Program in Natural and Synthetic Bioactive Products, Federal University of Paraíba, João Pessoa 58051-900, Brazil; fernandoferreira_15@hotmail.com (F.F.L.); nataliafsousa@ltf.ufpb.br (N.F.d.S.); hanrygb@hotmail.com (B.H.M.d.O.); denisecaiana@yahoo.com.br (M.D.L.F.); mtscotti@ccae.ufpb.br (M.T.S.); barbosa.ufpb@gmail.com (J.M.B.F.); luciana.scotti@gmail.com (L.S.); 2Post-Graduate Program in Development and Innovation of Drugs and Medicines, Federal University of Paraíba, João Pessoa 58051-900, Brazil; gabriellydduarte@hotmail.com (G.D.D.);; 3Post-Graduate Program in Pharmaceuticals Sciences Paraiba State University, Campina Grande 58429-500, Brazil; ricardo.olimpiodemoura@servidor.uepb.edu.br; 4Laboratory of Synthesis and Drug Delivery, Department of Biological Science, Paraiba State University, João Pessoa 58071-160, Brazil

**Keywords:** chalcones, anticancer activity, in vitro, drug discovery, natural products, synthesis

## Abstract

Chalcones are direct precursors in the biosynthesis of flavonoids. They have an α,β-unsaturated carbonyl system which gives them broad biological properties. Among the biological properties exerted by chalcones, their ability to suppress tumors stands out, in addition to their low toxicity. In this perspective, the present work explores the role of natural and synthetic chalcones and their anticancer activity in vitro reported in the last four years from 2019 to 2023. Moreover, we carried out a partial least square (PLS) analysis of the biologic data reported for colon adenocarcinoma lineage HCT-116. Information was obtained from the Web of Science database. Our in silico analysis identified that the presence of polar radicals such as hydroxyl and methoxyl contributed to the anticancer activity of chalcones derivatives. We hope that the data presented in this work will help researchers to develop effective drugs to inhibit colon adenocarcinoma in future works.

## 1. Introduction

Chalcones are direct precursors in the biosynthesis of flavonoids. They have an α,β-unsaturated carbonyl system (Figure 1), which gives them broad biological properties [1]. 

Their privileged structure also opens up possibilities for substitutions in aromatic rings [2], directly impacting its biological activity and development of formulations capable of improving its pharmacokinetic characteristics [3]. Due to their high biological value, the literature also reports on the anti-inflammatory [4], antiviral [5], antimicrobial [6], anticancer [7], antioxidant [8], and antifungal [9] properties of these compounds.

Cancer is generally characterized by the uncontrolled growth of cells, resulting from a regulatory dysfunction, which can be caused by several factors, both hereditary and environmental [10]. In addition to presenting a difficult prognosis, it still has a very aggressive treatment and a high mortality rate, caused both by the disease and by the impacts of its treatment [11]. Natural products such as chalcones of the type (1,3-diaryl-2-propen-1-ones) have been the focus of research for the treatment of this disease, as they have a conjugated carbonyl system that acts by inhibiting the polymerization of tubulin in tumor cells, interrupting its disordered reproduction cycle [12].

Colorectal cancer can be cited as an example, since the fatality rate is reduced by the removal of intestinal polyps through surgical procedures; however, in more advanced stages the surgery becomes ineffective, requiring the use of alternative therapies such as chemotherapy. Thus, a molecular target therapy becomes efficient because it has selectivity in abnormal cells [13].

According to the WHO (World Health Organization), in 2018 alone, cancer was responsible for more than 9.6 million deaths globally. Leukemia accounts for one in three cases of cancer, and is the most common type in children and adolescents (WHO, 2018) [14,15]. In women, breast cancer is the most common type found, corresponding to 25% of reported cases. The chemotherapy choice for this type of tumor aims to reach specific targets, usually using monoclonal antibodies such as trastuzumab [16]. New candidates derived from chalcones such as benzocoumarin-chalcones (Figure 2) have shown promise against breast tumors, as they are capable of inhibiting ERα and ERβ receptors, preventing the proliferation of abnormal cells [17].

Among these activities, studies of anticancer activity involving chalcones have been widely explored in recent years due to the difficulty in treating multidrug-resistant tumors with traditional medicines [18] in addition to the high toxic loads due to the association of drugs to combat the disease.

Chalcones with anticancer activity already known in the literature, such as MIPP and MOMIPP (Figure 3), act in the induction of methuosis in vitro, a type of non-apoptotic programmed cell death, while other chalcones act in different ways in the dysfunction of cell metabolism, causing death cell by apoptosis [12].

Given the importance of researching alternative treatments for different types of cancer, the present work focused on the report of natural and synthetic products based on the chalcone scaffold as potential candidates for the treatment of the most diverse types of cancer. The review was combined with a partial least square (PLS) analysis of the biologic data obtained from previous authors and the chemical structures with reported inhibitory activity of the HCT-116 lineage of colon adenocarcinoma, in order to identify descriptors that favor the anticancer activity of the compounds.

## 2. Review Approaches

The present review was written with the aim of presenting the anticancer activities of chalcones’ synthetic derivatives, aiming to demonstrate the physicochemical characteristics of these compounds that can contribute to their activity; however, it must be noted that this theme presents literature review papers that were previously published and which present different approaches and objectives that should be mentioned. The following types of approach were observed: (1)The first type refers to the description of the complete action mechanism of chalcones with emphasis on the biological targets and signaling pathways involved. Ouyang et al. (2021) [19] carried out a summary of recent advances in compounds belonging to the class of chalcones as potential anticancer agents and reported on the action mechanisms of these compounds. Furthermore, the authors presented future applications and scope of the chalcone family for cancer treatment and prevention. It is important to mention that in this work the authors emphasized the need for a complete description of the toxicity of chalcones. Also, it was possible to observe that the authors mentioned chalcones of natural origin, but emphasized the importance of the anticancer activity of chalcone hybrid compounds, mentioning the biological activity of artemisinin-chalcone hybrids, chalcone-azole hybrids, chalcone-coumarin hybrids, and chalcone-indole hybrids. In addition, the biological activity was addressed by describing the mechanism of action emphasizing the pathways responsible for anti-inflammatory activity, inhibition of MDR (multidrug resistance) channels, and anti-angiogenic, apoptotic, and tubulin polymerization. The authors emphasized the excellent activity of synthetic derivatives such as sofalcone (antiulcer agent) and methoxychalcone (choleretic drug), which both represent a promising strategy for developing chalcones as new anticancer agents. Another point mentioned referred to the lack of studies on chalcones in natural marine products. Thus, the authors concluded that chalcones are easy to synthesize, as well as easy to chemically modify, being able to generate compounds with a wide variety and structural diversity.(2)The second approach observed mainly emphasized synthetic compounds with the analysis of the chemical group related to activity, characterizing an approach of structure and activity approach. Among these articles, the main objective of the work developed by Shukla et al. (2021) [20] was to analyze the antitumor activity of chalcones 1,3-diaryl-2-propen-1-one through different mechanisms. The chemical groups related to activity and mentioned as being of greater importance referred to chalcone analogues with electron donating groups, indolyl, quinolone, pyrazolol, hydroxyaminobenzamide, hydroxamic acid, and pyridyl-indole groups. Among the activities evaluated were mechanisms related to apoptosis (emphasizing the importance of mitochondrial pathways), microtubule binding and cell cycle regulation, and inhibition of drug-metabolizing enzymes and new signaling pathways such as Notch (cell-surface Notch receiver). Thus, the authors concluded that the presence of electron donating groups such as OCH_3_, OH, halogens in the ring A or B of chalcones, induce apoptosis through the intrinsic or extrinsic pathway as they stabilize the enzyme-inhibitor complex through electrostatic interactions. Several chalcones with indolyl, quinolone, and pyrazole act as potential anti-thymicrotubule agents and interrupt the cell cycle, mainly in the G2/M phase. Apoptosis by enzyme inhibition is achieved by hydroxyaminobenzamide, hydroxamic acid groups, and pyridylindole.(3)The third approach found refers to synthetic chalcones and the reactions involved in their production. An article developed by Mastachi-Loza et al. (2021) [21] also emphasizes synthetic chalcones, but the authors mention cycloaddition mechanisms [4+2], which correspond to cycloadditions between a diene and a dienophile. The authors concluded that chalcones have not only been used as precursors of natural and synthetic molecules, but also in the development of new protocols and catalysts for asymmetric Diels–Alder nullification, reflecting their versatility. Furthermore, it is also possible that chalcones behave like dienes in Diels–Alder cycloadditions with inverse electron demand kinetics, as well as formal [4+2] cycloadditions, which are Diels–Alder cancellations. Therefore, due to their dual role in Diels–Alder cycloadditions, chalcones have wide possibilities in organic synthesis. Rani et al. (2019) [22] also addressed synthetic and conjugated chalcones, but presented the addition of a relationship and activity study as a differential, with an emphasis on their mechanism of action and anchorage studies along with their future therapeutic applications.(4)The fourth approach deals with the elucidation of the mechanism and performance of in vivo studies. This form of work was observed in the review conducted by Souza et al. (2021) [23], in which the authors aimed to describe the anticancer potential of chalcones related to some of the characteristics of cancer with emphasis on sustaining proliferative signaling, tumor-promoting inflammation, activation of invasion and metastasis, induction of angiogenesis and resistance to cell death; however, in this work no emphasis was shown on the chemical structure–the discussion was made in relation to the mechanism under study, and the chalcones which stood out were natural chalcones such as flavokawain B. Thus, the authors concluded that the chalcones and their derivatives had an anticancer effect by acting on the tumor microenvironment.

The following sections demonstrate articles that address the anticancer activity of chalcones.

## 3. Chalcones with Anticancer Activity

Chalcones of natural origin present a pattern of phenolic hydroxyls that originate from the biosynthetic reactions of flavonoids. As chalcones are metabolites derived from the mixed pathway, all natural chalcones have phenolic hydroxyls at positions 5 and 7 [24]. According to Leonte (2021) [25], this class of chalcones is very important, as they are crucial precursors in the biosynthesis of several metabolites with potential antitumor activity, such as flavones, flavanones, aurones, pyrazolines, pyrazoles, and epoxides.

Among the most used synthetic routes for the production of chalcones, the main route is through the Claisen–Schmidt condensation, which involves a condensation between a benzaldehyde and an acetophenone. They also have a conjugated ketone system, which can be functionalized through chemical reactions such as, for example, the addition of thiazole groups, producing a range of chalcone derivatives [25,26,27].

### 3.1. TNF-α

The TNF-α factor was discovered in the 1970s, and the function of a serum mediation of innate immunity that is responsible for the induction of hemorrhagic necrosis in tumors is attributed to this signaling pathway [28]. After a few years of studies, it was noticeable that this factor has a dual action in cancer, especially in breast cancer, in which TNF-α can be a target that causes the disease as well as a therapeutic agent [29].

Articles that address TNF-α and natural chalcones correspond to research carried out by Roh and collaborators (2020) [30], which addressed the structural optimization of natural chalcones: isoliquiritigenin, which corresponds to a trihydroxy chalcone (1), as well as butein (2), which represents a tetrahydroxychalcone. Compounds that have already been reported as good inhibitors of histone deacetylases (HDAC) [31] are also considered to be inhibitors of this inflammatory mediator, but their physical-chemical characteristic of solubility is inconvenient for their study and use, because despite having polar groups, these compounds have low solubility (79 mM, 21.0 mg/mL) and a partition coefficient (Log P) of only 0.42, in addition to demonstrating low potency (IC50, 43.3 mM) and insufficient efficacy, with only 21% inhibition at 20 mM in vitro. Due to these limitations, the authors carried out pro-drug optimization strategies by conducting structural modifications in the two mentioned chalcones, with the main objective of improving their pharmacokinetic properties. Based on this information, derivatives of butein were synthesized, corresponding to compounds **3**, **4**, and **5** (Figure 4), which showed the power to suppress up to 50% of TNF-α production in peritoneal macrophages of mice after stimulation with lipopolysaccharides. Compound **5** was shown to be the most potent inhibitor, with an in vitro IC50 of 14.6 mM and limb volume suppressed by 70% in a murine lymphedema model. Thus, the authors concluded that the pro-drug strategy allowed a six-fold increase in the kinetic solubility of compound **1** and five-fold higher levels of the active metabolite in the blood for compound **5** with oral administration in the pharmacokinetic study. When undergoing modifications in their structure in order to facilitate solubility and permeability, an increase in potency was equivalent to three times (17.3 mM) for anti-inflammatory effects in vitro than the unmodified compound (43.3 mM). Compound **4**, despite its insufficient solubility (28 mg/mL), showed greater permeability and was able to suppress limb swelling by 53% orally (100 mg/kg/day). Compound **5** was the most potent (14.6 mM) and had five times greater solubility (136 mg/mL). As a pro-drug, compound **5** was rapidly converted to compound **2** by liver microsomes of three species, including mice, rats, and humans. Thus, the authors suggested that compound **5** could be developed as a potential therapeutic agent targeting anti-inflammatory activity to alleviate the progression of lymphedema.

In a study carried out in 2019, Hashid and collaborators [32] observed that chalcones that have hydroxy, methoxy, and chlorine groups in the ortho and para positions, have strong inhibitory actions against mechanisms of inflammatory action, such as inhibition of cyclooxygenase. It was also revealed in this study that treatment with these compounds was able to inhibit up to 90% of inflammatory edemas.

### 3.2. Colon Cancer

As potential therapeutic agents for colon cancer, articles that reported synthetic chalcones were observed with chalcones-ciprofloxacins linked to 1,2,3-triazole (6) (Figure 5), which showed an IC50 ranging from 2.53–8.67 mM, 8.67–62.47 mM, and 4.19–24.37 mM for HCT116, HT29, and Caco-2 cell lines, respectively; while doxorubicin showed IC50 values of 1.22, 0.88, and 4.15 mM. In addition, the compounds in studies still showed Topoisomerase I and II inhibitory activity [33]. Based on the properties of anthocyanidins and aglycones, Păușescu and collaborators (2022) [34] carried out a synthesis of derivatives of the flavilium cation (7) (Figure 5), these being evaluated for anticancer activities in HCT116 and HepG2 strains. The anticancer effect was influenced by the position (6-, 7-, or 8-) of the methoxy group on the β-ring for the methoxy-4′-hydroxy-3′ methoxyflavilyl cation. Thus, the authors concluded that the evaluation of the anticancer activities of derivatives containing methoxy groups in the flavilium cation in hepatocellular carcinoma cells (HepG2) and colon cells of the HCT116 lineage, showed greater efficiency even at low concentrations (26 µM). Formation of the inclusion complex of Compound **5** with the cyclodextrin derivative SBECD led to a 1.5-fold increase in water solubility, preserving 70% of the cytotoxic effect in HepG2 cells.

In a study conducted by Palko-Labuz and collaborators (2020) [35], the natural chalcone found in the Kawa plant, flavokawain B (3) was evaluated for its inhibitory capacity of LoVo/Dx cells. The results showed that the natural chalcone under study showed strong cytotoxic activity, as well as strongly inhibited cell proliferation of the strain under study. Furthermore, at low concentrations the chalcone flavokawain B (8) contributed to apoptosis, as it led to an increase in the expression of caspase-3 activity.

Vanillin-based chalcone analogues were discussed by Lukovic and collaborators (2020) [36]. The IC50 values observed in the HCT-116 cell models were equivalent to 6.85 ± 0.71 μg/mL for Compound **9** (Figure 5) and 7.9 ± 1.37 μg/mL for Compound **10** (Figure 5). Furthermore, vanillin-based chalcone analogues caused overexpression and activation of mitochondrial Bax protein and caspase-3 in HCT-116 cells, indicating that their antitumor mechanism of action was mediated by activation of the internal apoptotic pathway.

Inhibition of HCT-116 cells for evaluation of new tubulin inhibitors was performed by Hamashi and collaborators (2021) [37], through the aldol reaction of N-tosyl imidazolketone with the respective aldehyde group (Compound **11**) (Figure 5), thus obtaining 26 compounds that obtained GI50 values corresponding to 5.14 ± 6.81 µM. Regarding the IC50 values of the enzyme histone deacetylase (HDAC), these were equivalent to 1.8 ± 9.0 µM.

The presence of groups that favor hydrogen bonds, as well as the presence of halogenated elements may be related to the activity, since among the compounds considered the most active, they have these characteristics in common. In contrast, these groups demonstrate large cytotoxic loads, as demonstrated by Palko-Lobuz and collaborators [35].

### 3.3. Lung Cancer

To obtain new therapeutic agents for lung cancer, Mphahlele and collaborators (2021) [38], carried out a sulfonylation reaction on compounds of the type 5-styryl-2-aminochalcones with p-toluenesulfonyl chloride in pyridine providing new hybrids of 5-styryl-2-sulfonamidochalcones. The in vitro results of the compounds (**12**) and (**13**) (Figure 6) against A549 and vero cell lines with LPS, showed a suppression capacity of up to 55% of ROS in vero cells and 35% in A549 cells. These compounds also reduced the cytotoxicity against the A549 cell line and did not affect the viability of vero cells. Another mechanism used for the discovery of drugs against lung cancer was addressed by Sherikar and collaborators (2021) [39]; in this work, the authors evaluated synthetic chalcone derivatives for blocking calcium channels, the study being carried out through pharmacophore modeling combined with experimental evaluation. Pharmacophore modeling revealed that hydrogen bonding receptors and hydrophobic groups are important features for calcium channel-blocking activity. The docking study showed the existence of hydrophobic interactions, hydrogen bonds, and Van der Waals interactions between the amino acid residues and the ligands. In vitro screening showed that compounds **14**, **15**, and **16** were potent, yielding an IC50 of 4756, 3608, and 5211 µM, respectively, while the standard drug, Nifedipine, showed an IC50 of 1.30 µM. Furthermore, it is important to mention that synthetic chalcone derivatives with NO (nitric oxide) donating capacity is promising for designing new calcium channel blockers.

Another class under study was approached by El-wakil and collaborators (2020) [40], which refers to a series of 1,3,5-triazines linked to chalcones. The results showed that compounds (**17**) and (**18**) (Figure 6) significantly inhibited the viability of cancer cells of the A549 lineage, and their IC50 values were 24.5 and 17 µM, respectively, in reference to cisplatin (IC50 = 21.5 µM). Furthermore, mechanistic studies employing MALDI-TOF MS (matrix-assisted laser desorption ionization—time of flight mass spectrometry) and fluorescence spectroscopy using the EvaGreen probe inferred that (17) and (18) induced DNA double-stranded breaks, in contrast to cisplatin, which induces crosslinks between DNA strands’ DNA.

The functionalization of coumarins to chalcones as potent agents for lung cancer, among other activities, was mentioned by Kumar and collaborators (2021) [41]. In this article, the authors synthesized coumaryl–chalcone derivatives and it was observed that the compound lead (**19**) (Figure 6) is very potent against the A-549 (lung) strain, and also showed satisfactory results for the Jurkat (leukemia) and MCF-7 (breast). The IC50 values were, respectively, equivalent to 70.90, 79.34, and 79.13 μg/mL.

The natural chalcone flavokawain B (Compound **20**) (Figure 6) was studied by Hseu and collaborators (2019) [42] as an inhibitor of A459 cells and NSCLC cells (H1299). IC50 values of 11 µg/mL were observed for A549 cells and 5.1 µg/mL for H1299 cells. Furthermore, at concentrations of 5–15 μg/mL, the chalcone under study induced apoptosis and autophagy in A-549 cells.

The platinum complex functionalization in the chalcone structure was a strategy adopted by Wang and collaborators (2021) [43]. Compound **21** (Figure 6) was the best performing complex, showing high apopytotic capacity, as well as IC50 values of 0.31 ± 0.09 and 0.71 ± 0.18 μM for the resistant strains A549/CDDP and SGC-7901/CDDP, while IC50 values for cisplatin corresponded to 35.05 ± 1.39 and 26.81 ± 1.73 μM, respectively.

Bischalcone derivatives linked to aliphatic ligands, with furan units in the A or B rings were synthesized by Fathi and collaborators (2021) [44]. Substances 22 and 23 (Figure 6) were considered the most promising, with IC50 (24.9 and 13.7 μg/mL, respectively) against A549, compared with the reference drug doxorubicin (IC50, 28.3 μg/mL), in addition to presenting IC50 (26.1 and 14.4 μg/mL, respectively) against A431 compared with the reference drug doxorubicin (IC50, 24.9 μg/mL).

A ligustrazine chalcone was synthesized by Bukhari (2022) [45]. Compound **24** (Figure 6) showed an IC50 of 5.11 µM, as well as an inhibitory potential of other strains such as MCF-7.

A study led by Gaur and collaborators [46] pointed out that compounds of the chalcone class which have indole and furan groups in their structure, present high cytotoxicity against cell lines of this specific type of cancer, capable of combating these cells with an IC50 of 1 µg/mL, as well as cyclized chalcones with groups of diazoles, as reported by Bracke et al., 2008 [47].

### 3.4. Breast Cancer

Breast cancer was one of the most mentioned pathologies, Wang and collaborators (2020) [48] reported the synthesis of new chalcones containing a diaryl ether moiety (25) (Figure 7). The results showed that among the synthesized compounds, the compound (**25**) with the 4-methoxy substitution on the right aromatic ring was considered the most active in cancer for MCF-7, HepG2, and HCT116 cell lines, showing IC50 values of 3.44 ± 0.19, 4.64 ± 0.23, and 6.31 ± 0.27 μM, respectively. In vitro tests showed that the compound (**25**) can effectively inhibit tubulin polymerization. Further studies of the mechanism of action revealed that the compound (**25**) was able to induce G2/M phase arrest and cell apoptosis. Furthermore, molecular docking studies revealed that compound (**25**) interacts and binds at the colchicine binding site of tubulin.

In another work Guruswamy and Jayarama (2020) [49] carried out synthesis of chalcone derivatives by the Claisen–Schmidt method. The resulting compound (**26**) (Figure 7) demonstrated incredible anticancer potential in MCF7 cells, with IC50 estimates of 6.55–10.14 µM. In addition, this induced apoptosis, since the increase in the expression level of Caspase 9 and Caspase 3 was noticeable. These results demonstrated expressive apoptotic activity.

The authors Homerin and collaborators (2020) [50], carried out synthesis of chalcones with thienyl groups as potential inhibitors of the farnesyltransferase enzyme as well as of the MCF-7 cell line. The bis(thienyl) chalcone (compound **27**) (Figure 7) was the most active compound in the series (IC50 = 7.4 µM), thus showing antiproliferative potential against the MCF7 cancer cell lines.

Another reported class refers to ethoxychalcones; these were studied by Harshitha and collaborators (2020) [51] and were obtained by classic aldol condensation. The best compound was derivative (**28**) (Figure 7), which showed IC50 value = 53.47 μM for breast cancer cell line MDA-MB-231 and metastatic melanoma cells (A-375).

Al-kaabi and collaborators (2021) [52] synthesized chalcones from 1,2-bis(2-methoxy-4-vinylphenoxy)ethane. Compounds (**29**), (**30**) and (**31**) (Figure 7) showed the highest inhibition rate against the human breast cancer cell line Cal51, 64.1% 60.2%, and 50.4%, respectively.

Chalcones with thiophene groups were synthesized by Mangoud and collaborators (2020) [53]. The results showed that chalcone 32 (Figure 7) had an inhibition percentage of 56.90% against T-47 D.

Polymethoxylated chalcones substituted with nitro groups (NO_2_) were synthesized by Ahn and collaborators (2022) [54]. IC50 values for the MCF-7 cell line were around 1.33 and 172.20 μM, with compound **33** (Figure 7) being considered the best performer.

### 3.5. Oral Cancer

In cases where the search for new therapeutic agents for oral cancer was mentioned, this was related to the suppression of inflammatory mediators such as interleukins. Rajeswari and collaborators (2022) [55], performed the functionalization of chalcones with other natural products belonging to the coumarin class, which are considered pharmacophoric groups and which have already reported activity against oral cancer. In addition, pyrazolone aldehydes were produced using the Vilsmeier–Haack reaction, via the reaction between aldehydes and ketones in an alcoholic medium with sodium hydroxide. The in vitro assays were combined with in silico simulations that demonstrated high affinity of the compounds under study with mediators such as interleukins, as it was also possible to observe that the in vitro cell viability studies of the series show that chalcones (34), (35), and (36) (Figure 8) presented IC50 values of 2.96, 2.97, and 2.82 µM against CAl27 oral cancer cell lines.

The inhibitory activity of the AW13516 cell line was studied by Kode and collaborators (2020) [56]. In this work, the authors synthesized the compound **37** (*E*)-1-(3,4-dimethoxyphenyl)-3-(1-methyl-5-(3,4,5-trimethoxybenzoyl)-1H-indol-3-yl)prop -2-in-1-one (37) (Figure 8). This derivative has 1-methyl, 2 and 3-methoxy substituents on the aromatic ring and was effective in inhibiting the AW13516 strain, showing GI50 values of 0.96 µM.

A study guided by Gul and collaborators [57] demonstrated that chalcone-like compounds have a greater selectivity for cells of the oral cancer lineage when they are trimethoxylated. In addition to the selectivity, based on in vitro experiments the authors concluded that these groups increase the potency of the compounds, showing promise in the fight against this type of cancer.

### 3.6. Leukemia

Leukemia was also one of the most prevalent pathologies; synthetic derivatives of chalcones were reported in research by Mphahlele and collaborators (2022) [58], a series of 2-hydroxy-3-nitrochalcones substituted by 5-methyl, 5-bromo, and 5-chloro were synthesized. The results showed that chalcones 38 to 45 (Figure 9) exhibited inhibitory effect against α-glucosidase and α-amylase enzymes, in addition, it exhibited minimal cytotoxicity against Raw-264.7 macrophage cells (murine) in comparison with the anticancer drug, curcumin. For acute lymphoblastic leukemia, Kudličková and collaborators (2020) [59] performed the synthesis of 20 chalcone derivatives substituted with nitro groups (NO_2_) and (CF3). Of the synthesized compounds, four derivatives (compounds **46**, **47**, **48**, and **49**), presented IC50 values between 6.1 and 8.9 μM for the inhibition of T lymphocytes, thus demonstrating a high antiproliferative role.

Del Rosário and collaborators (2022) [60], carried out the synthesis of 50 chalcones through a standard aldol condensation reaction of three acetophenones with three benzaldehydes. These compounds were evaluated by means of the inhibition capacity of the strains HL-60 and U-937. Chalcone 2,2′-furoyloxy-4-methoxychalcone (compound **50**) (Figure 9) was the most active compound, with IC50 values corresponding to 4.9 ± 1.3 μM.

In a differentiated methodology, Li and collaborators (2021) [61] aimed to search for new inhibitors of the specific histone lysine demethylase 1 (LSD1) enzyme, for which the synthesis was carried out through the aldol condensation of acetophenone and benzaldehyde. Compound (**51**) (Figure 9), characterized as a piperidine oxazole chalcone, presented IC50 values corresponding to 0.14 μM, thus being about 100 times more potent than its precursor, in addition to representing a highly promising compound for the treatment of leukemia.

Petrov and collaborators (2020) [62], carried out the synthesis of benzoxazolone derivatives to evaluate the inhibition of human leukemia strains. Compound (**52**) (Figure 9) demonstrated dose-dependent effect of cytotoxicity, being more sensitive for BV-173, SKW-3 and HL-60 strains (IC50 = 3.6–10.7 μM). The introduction of the aminomethyl group at position 3 of benzoxazolone was considered a structural prerequisite for the cytotoxic activity of the synthesized molecules.

For acute lymphoblastic leukemia, Kudličková and collaborators (2021) [59] performed the synthesis of 20 chalcone derivatives substituted with nitro groups (NO_2_) and (CF3). Of the synthesized compounds, four derivatives (compounds **46**, **47**, **48**, and **49**), presented IC50 values between 6.1 and 8.9 μM for the inhibition of T lymphocytes, thus demonstrating a high antiproliferative role. The greatest efficacy was demonstrated by chalcones against Jurkat leukemic cells, which are rapidly proliferative and more sensitive cells. IC50 values (excluding three compounds) ranged from 3.9 to 15 µM. The best results were obtained by compound **53** (Figure 9).

Bis-quinolinyl-chalcone compounds were synthesized by Insuasty and collaborators (2020) [63]. Compound **54** (Figure 9) showed significant activity against leukemia cells K-562 (GI50 = 0.88 µM), RPMI-8226 (GI50 = 0.32 µM) and SR (GI50 = 0.32 µM).

Studies led by Vrontaki (2017) [64] and Mercader (2012) [65] showed that nitrogenated and halogenated chalcones have high cytotoxicity against leukemic cells, being able to limit their growth, in addition to inducing the process of cell apoptosis.

### 3.7. Hepatocarcinoma

A study conducted by Wang and collaborators (2022) [66] showed two series of chalcone derivatives containing aminoguanidine or bis-chalcone that were designed, synthesized, and screened for their cytotoxicity, proliferation inhibition, and apoptosis-promoting activity in vitro. The results showed that 2-((*E*)-4-((*E*)-3-oxo-3-(p-tolyl)prop-1-en-1-yl)benzylidene)hydrazine-1-carboximidamide (58) (Figure 10) was the most potent compound, with IC50 values of 7.17 μM and 3.05 μM of in vitro anti-proliferative activity against HepG2 human hepatocarcinoma cells and SMMC-7721 cells, respectively. This result showed that the compound had a certain degree of selectivity for human hepatocellular carcinoma cells, especially for SMMC-7721, affirming this compound as a potential drug candidate.

Helmy and collaborators (2022) [2], synthesized 4-acetyl-5-furan/thiophene-pyrazole derivatives. Compound (**59**) (Figure 10) showed to be the most promising compound, with IC50 = 26.6 µg/mL against HepG2 cells compared with the reference drug doxorubicin (IC50 = 21.6 µg/mL), and with IC50 = 27.7 µg/mL against A549 cells compared with the reference drug doxorubicin (IC50 = 28.3 µg/mL).

Huang and collaborators (2020) [67], reported functionalized platinum complexes to chalcones. Chalcone 60 (Figure 10) showed the best IC50 value for the HepG-2 strain IC50 of 0.33 μM, as well as other good values 0.41 μM, 0.30 μM, 0.45 μM, and 11.85 µM for HeLa, MGC-803, NCI-H460, and HL-7702, respectively.

α-phthalimido-chalcones were synthesized by Mourad and collaborators (2020) [68]. Trimethoxy derivative 61 (Figure 10) demonstrated the most potent anticancer activity, with an IC50 of 1.62 µM for Hep G2 and 1.88 µM for MCF-7.

Thiophenic, nitrogenous, and diazolic chalcones have a strong antioxidant power, in addition to acting significantly against breast cancer and hepatocellular carcinoma cells, as reported in studies led by Zahrani in 2020 [69].

### 3.8. Cervical Cancer

Methoxylation and hydroxylation reactions in chalcones were the strategies used by Sangpheak and collaborators (2019) [70] for the elaboration of new topoisomerase enzyme inhibitors. The synthesized compound corresponded to a derivative that had 2,4-dimethoxy and 6-hydroxy groups in ring A and 3′,4′,5′-trimethoxy in ring B (Compound **62**) (Figure 11), and showed the highest cytotoxicity in in vitro against HeLa, HT-1376, and MCF-7 strains, whose IC50 values corresponded to 3.2, 10.8, and 21.1 µM. In addition, it was demonstrated that the test compound had a high affinity for the topoisomerase enzyme, which was confirmed by molecular docking and molecular dynamics simulations.

HeLa cell inhibition was evaluated by Vongdeth and collaborators (2019) [71]. In this work, polyhydroxychalcones were synthesized by the classic Claisen–Schmidt condensation pathway of 2-hydroxy-4,6-dimethoxyacetophenone with several aldehydes. Compound (**63**) (Figure 11) was the most potent, with greater selectivity for HeLa cells (IC50 1.44 µM) and SK-OV-3 cells (IC50 1.60 µM).

### 3.9. Glioblastoma

Medanha and collaborators (2021) [72] reported the synthesis of new chalcones as promising inhibitors of human glioblastoma lines U98 and GL261. Chalcone 65 (Figure 12) was the compound with the best performance and reduced cell proliferation by 40% for U87 cells and 25% for GL261, not dependent on exposure time. The number of viable cells was significantly lower in treated cells and the invasive capacity of U87 cells was reduced by 50% after treatment with chalcone 65. These results demonstrate chalcone 65 as a promising agent.

Sulfonamide groups in chalcones were reported by Custodio and collaborators (2020) [73]. Compound (**66**) (Figure 12) had the lowest values of IC50 (2.1 e 2.4 µg.mL^−1^ against SF-295 e PC-3, respectively).

### 3.10. Melanoma

Castano and collaborators (2022) [74], synthesized chalcone-sulfonamide hybrids. Since the compound (**67**) (Figure 13) presented the best inhibition profile for LOX IMVI (melanoma) with IC50 = 0.34 µM, it also showed good results for MCF7 and MDA-MB-468 (breast cancer) with IC50 values of 0.97 and 1.20 µM, respectively; K-562 (leukemia) with IC50 = 1.50 µM, and HCT-116 (colon cancer) with IC50 = 1.49 µM.

## 4. Other Studies

This section includes compounds that are derived from chalcones, but have a different structure, as well as other mediators of anticancer activity. The results are described in Table 1.

## 5. Chemometric Analyses

The PLS analysis was performed with the training set of 128 descriptors, assigning classification criteria to the compounds according to the activity value of the compounds obtained in the literature review. The activity values included were equivalent to the IC50, GI50, or percentage of inhibition (%) obtained for the cell lines under study. The selection of compounds related to the HCT-116 lineage of colon adenocarcinoma was carried out. This series corresponded to 28 compounds that presented the activity value in IC50 (µg/mL). The total series under study was divided into active and inactive compounds, with a value of +1 for active compounds and -1 for inactive compounds. Compounds **01**, **02**, **03**, **04**, **05**, **06**, **07**, **08**, **09**, **11**, **12**, **14**, **22**, **27**, and **28** were classified as active, as they presented IC50 values < 10 µg/mL (Figure 14A). The other compounds in the model, referring to compounds with IC50 values > 10 µg/mL, were classified as inactive (Figure 14B). To improve the statistical validation of the model, compounds **14** and **22** were excluded. The model validation was performed with the leave-one-out (LOO) cross-validation correlation coefficient method, resulting in a good observed statistical index in the calculation of the latent variable LV5, having a good predictive coefficient (Q^2^ = 0.847) and an excellent determination coefficient (R^2^ = 0.960), reinforcing the quality of the physical-chemical descriptors of VolSurf and the biological activity data used in this study.

The t1-t2 PLS Score plot is shown in Figure 15. Regarding the plot, the selected model provides good discrimination between active and inactive class of compounds according to the statistically significant quality of the derived PLS model.

Based on the developed model, the contribution coefficients of the most significant descriptors for the activity were analyzed, that is, we sought to highlight the descriptors that favor the biological activity of the active compounds and the descriptors that disfavor the biological activity of the compounds. Among these descriptors, it is possible to observe the positive influence of the descriptors Flex, G, W1, and Pol while the negative influence can be highlighted by the MetSab descriptor, as shown in Figure 16.

The mentioned descriptors take into account the structural characteristics of the compounds without declaring the interference of biological activity. As positive contributions, the following stand out:-The W1 descriptor belongs to the hydrophilic volume descriptor block, which describes the accessible molecular envelope that interacts attractively with water molecules.-The FLEX descriptor is related to flexibility parameters and represents the maximum flexibility of a molecule.-Another descriptor is the POL, which represents an estimate of the average molecular polarizability and is based on the structure of the compounds.-Molecular globularity (G) is related to molecular flexibility and is defined as S/Sequiv with Sequiv = surface area of a sphere of volume V, where S and V are the molecular surface and volume described above, respectively.

As negative contributions, the following stand out:

The MetSab descriptor is an ADMET (absorption, distribution, metabolism, excretion, toxicity) descriptor, which represents the metabolic stability after incubation with the human CYP3A4 enzyme.

So in this way, it is understood that the series under study has hydrophilic characteristics and may present metabolization inconveniently.

## 6. Discussion

According to Michalkova and collaborators (2021) [116], chalcones are molecules chemically derived from aromatic ketones, and their chemical structure is considered simple, which may explain the ease of modifications in their structure. This information can be confirmed in the analyzed data, since most of the articles found referred to obtaining synthetic or semi-synthetic chalcones. The only natural chalcones mentioned referred to the butein and isoliquiritigenin derivatives and the chalcone present in the Kawa plant, known as flavokawain B.

Regarding the semi-synthetic chalcones that were observed, the most used synthesis reaction corresponded to the Claisen condensation, which is configured as a standard in obtaining chalcones, however, it was noticeable that the structural modifications carried out referred to hydroxylation reactions and methoxylation. The addition of halogens such as chlorine, bromine, and fluorine was also observed. In addition, the high occurrence of thiophene rings and azole substituents was noticeable. Another strategy used refers to the elaboration of complexes with platinum.

It was observed that the addition of the metal complex to the chalcone structure significantly increased the potency of the chalcone under study, being noticeable by the decrease in the IC50 values. According to Hacker (2009) [117] platinum complexes such as cisplatin and carboplatin demonstrate their antitumor action through the formation of DNA adducts that consequently contribute to the inhibition of DNA replication and transcription.

The high use of the strategy of adding polar radicals such as the hydroxyl and methoxyl was identified in the analyses related to the PLS regression, since the descriptors that are most related to the activity of the compounds under study are descriptors related to polar features, such as the W1 descriptor and the G descriptor.

Among the types of strains under study, the strains related to lung cancer were the most prevalent, reported in nine studies with the A-549 strain being the most mentioned strain. In addition, it was also observed that the compounds in question have multi-target potential since, in several cases, it was observed that natural chalcones and chalcone derivatives under study showed inhibitory capacity in more than one cell lineage, such as the natural chalcone flavokawain B.

Based on the observed results, the class of chalcones can be confirmed as potent anticancer agents, as well as their low level of toxicity and ease of procurement.

## 7. Material and Methods

### 7.1. Literature Review

A literature review was carried out with articles that addressed the anticancer activity of natural chalcones, synthetic chalcones, and chalcone derivatives. The search was carried out on the Web of Sciences database (https://www.webofscience.com/wos/woscc/basic-search (accessed on 10 January 2023)); the period comprised the last four years, from 2019 to 2023, and the words used as descriptors were chalcones, anticancer activity, and in vitro. The inclusion criteria were full articles published in English that were reporting anticancer activity of natural or synthetic chalcones and chalcone derivatives. The exclusion criteria were in vivo and in silico studies, chalcone analogs, and other biological activities.

In the search were found 159 articles from the Web of Science database, but only 101 met the review eligibility criteria mentioned above.

The SMILES referring to all the compounds reported in the bibliographic survey are described in Appendix A.

### 7.2. Data Set

The chemical structures of the compounds under study were drawn in the software Marvin Sketch 22.13., 2022, ChemAxon (https://chemaxon.com/ (accessed on 10 January 2023)) [118]. After drawing, the structures were converted into SMILES, which consist of the canonical representation of the structure.

### 7.3. Chemometric Studies

The structures in three dimensions (3D) saved in SDF format were imported in the program VolSurf+ v.1.0.7 [119] and subjected to molecular interaction fields (MIF) to generate descriptors using the following probes: N1 (nitrogen-amide binding donor probe—N1); O (carbonyl oxygen-hydrogen bond receptor probe); OH2 (water probe) and a DRY (hydrophobic probe). Non-MIF-derived descriptors were generated to create 128 descriptors [120].

### 7.4. Partial Least Squares (PLS)

The QSAR analysis with PLS involves working with matrices X and Y, respectively, corresponding to chemical variables and biological descriptors. The PLS method approximates the X and Y matrices to model the relationship between them. The PLS methodology was applied to build models. A discriminant PLS analysis was performed, with active compounds assigned the value of 1 (totaling 14 compounds) and inactive compounds assigned the value of −1 (totaling 14 compounds) separated according to the values of the IC50 (µg/mL), where the compounds that showed IC50 values < 10 µg/mL were considered active and the molecules that showed IC50 values > 10 µg/mL were considered inactive. The number of latent variables (LV) as well as the original value of the variables was selected using a graph of coefficients together with a graph that responds for each PLS component; the variation explained in the inflow (R^2^) and the variation explained in forecast (Q^2^) cv; cross-validation performed by LOO. The model with the highest cross-validation correlation based on the coefficient value (Q^2^ cv) was selected.

## 8. Conclusions

Based on the literature review, it is possible to state that the class of chalcones that presents inhibitory activity against several cancer strains can be attributed to natural and synthetic chalcones. In addition, it was observed that the chalcone flavokawain B showed a high potential for cell inhibition in two cell lines, presenting multi-target potential. Another important point is the low toxicity reported in this class of compounds.

## Figures and Tables

**Figure 1 molecules-28-04009-f001:**
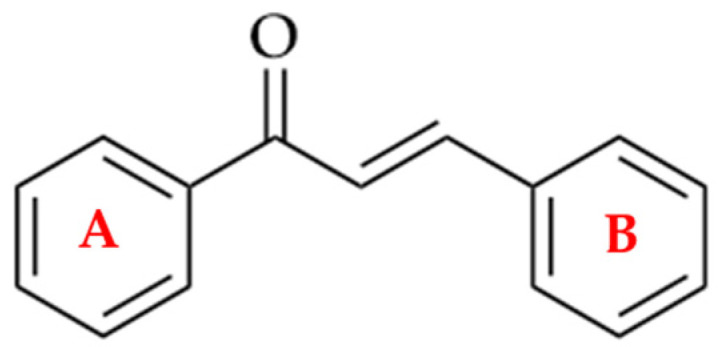
Chemical structure of a chalcone.

**Figure 2 molecules-28-04009-f002:**
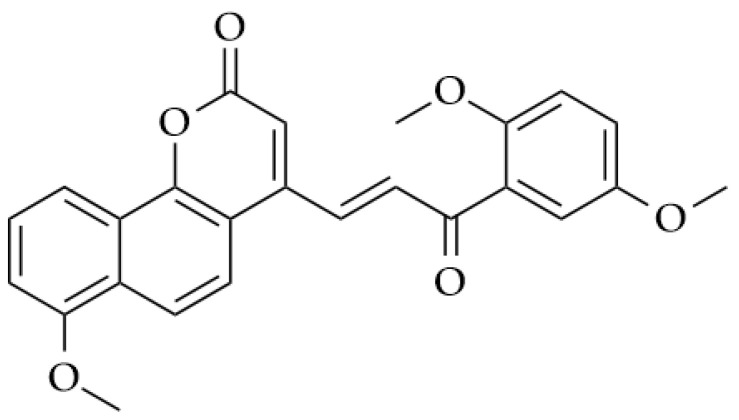
Chemical structure of benzocoumarin-chalcones.

**Figure 3 molecules-28-04009-f003:**
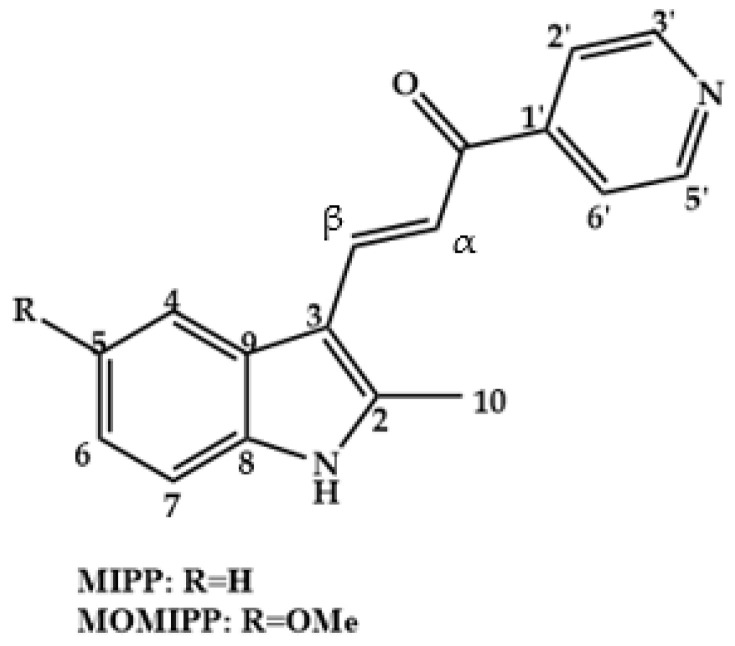
Chemical structure of chalcones, MIPP and MOMIPP.

**Figure 4 molecules-28-04009-f004:**
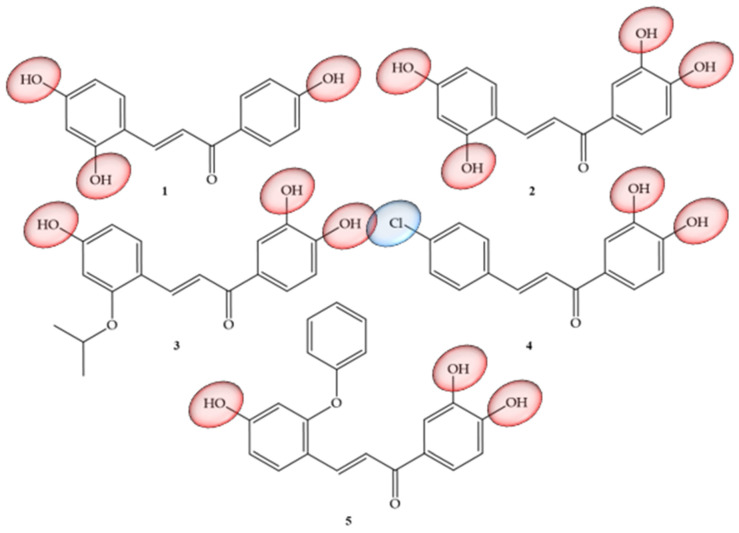
Chemical structure of TNF-α inhibitor compounds.

**Figure 5 molecules-28-04009-f005:**
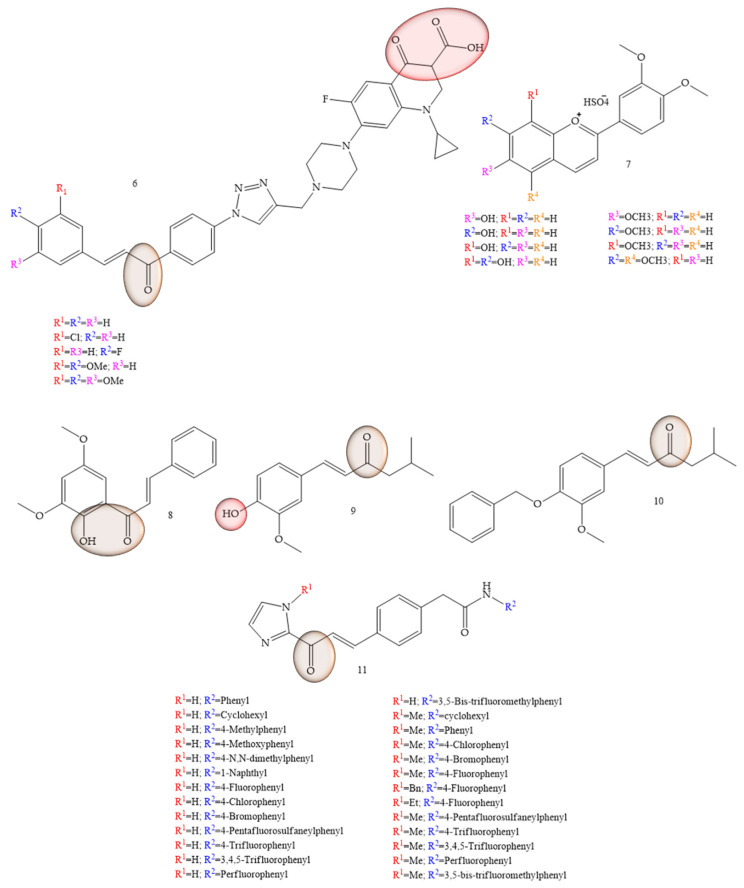
Chemical structure of colon cancer inhibitor compounds.

**Figure 6 molecules-28-04009-f006:**
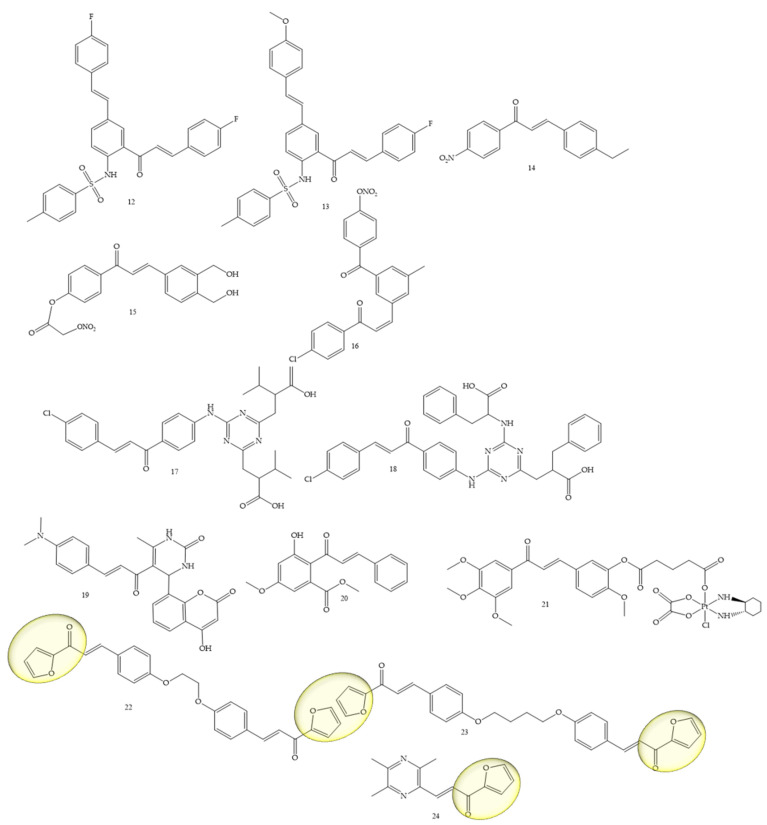
Chemical structure of lung cancer inhibitor compounds.

**Figure 7 molecules-28-04009-f007:**
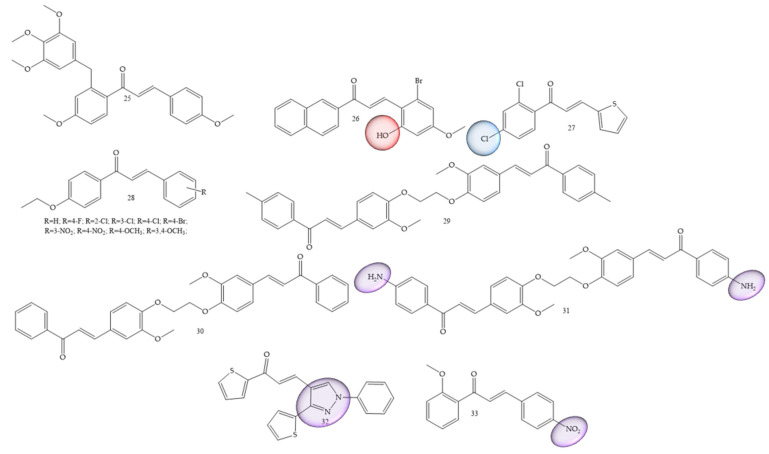
Chemical structure of breast cancer inhibitor compounds.

**Figure 8 molecules-28-04009-f008:**
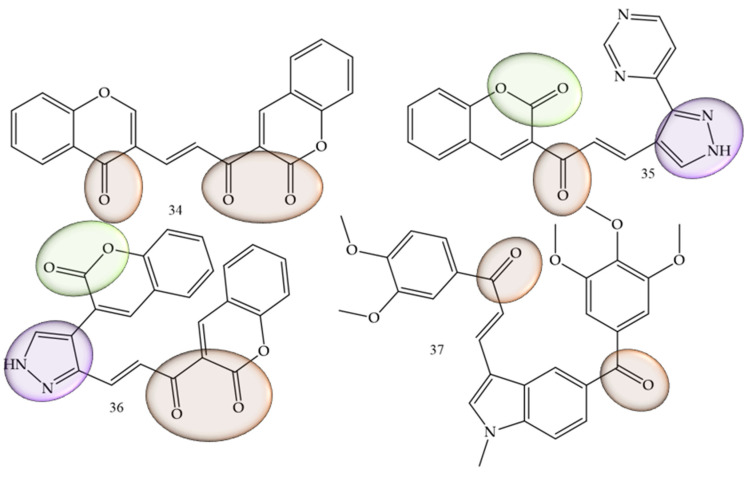
Chemical structure of oral cancer inhibitor compounds.

**Figure 9 molecules-28-04009-f009:**
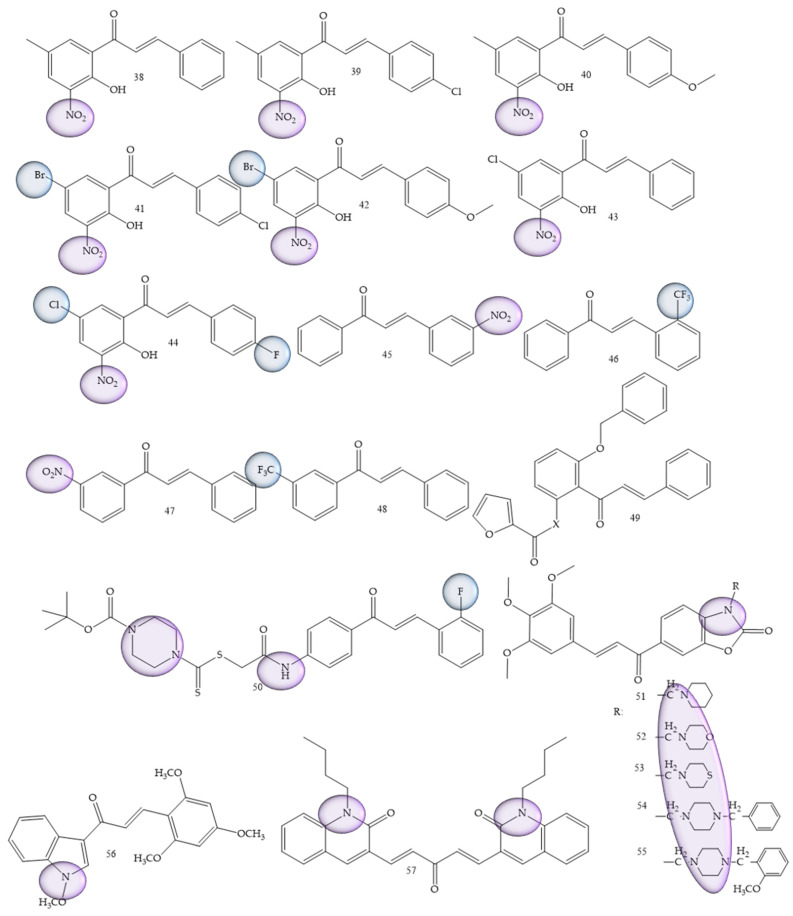
Chemical structure of leukemia inhibitor compounds.

**Figure 10 molecules-28-04009-f010:**
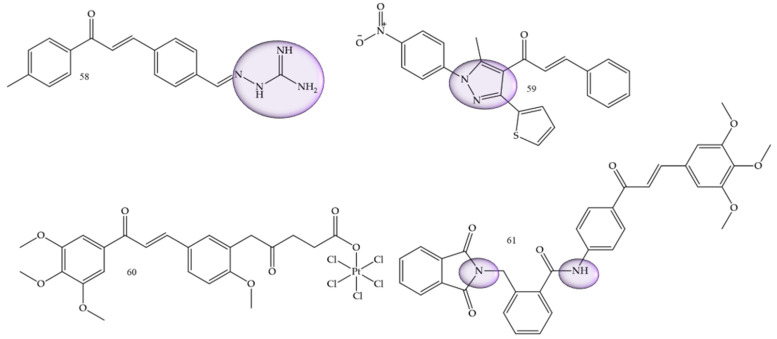
Chemical structure of hepatocarcinoma inhibitor compounds.

**Figure 11 molecules-28-04009-f011:**
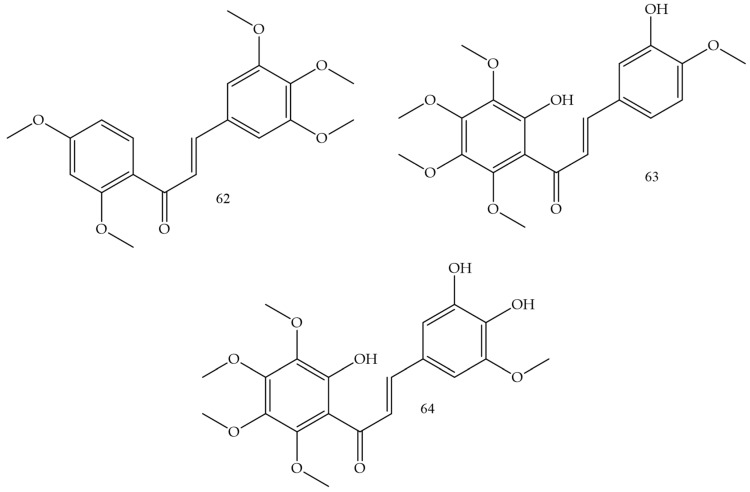
Chemical structure of cervical cancer inhibitor compounds.

**Figure 12 molecules-28-04009-f012:**
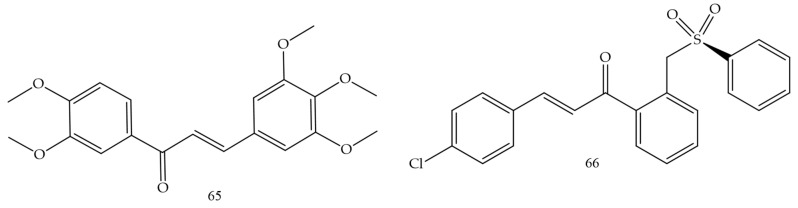
Chemical structure of glioblastoma inhibitor compounds.

**Figure 13 molecules-28-04009-f013:**
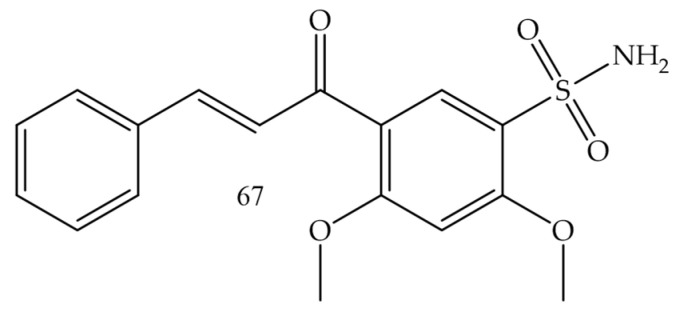
Chemical structure of melanoma inhibitor compounds.

**Figure 14 molecules-28-04009-f014:**
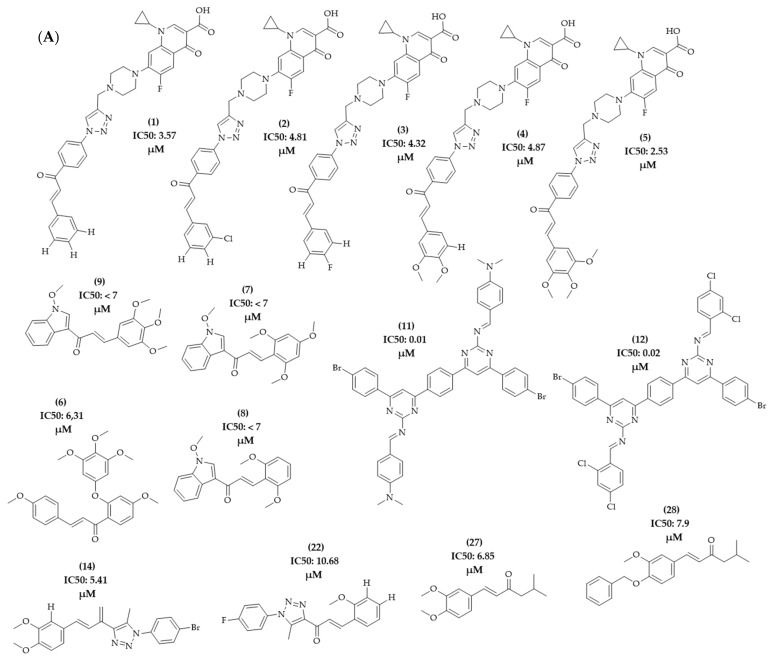
Chemical structure of the series of compounds submitted to PLS analysis. (**A**) Active compounds; (**B**) inactive compounds.

**Figure 15 molecules-28-04009-f015:**
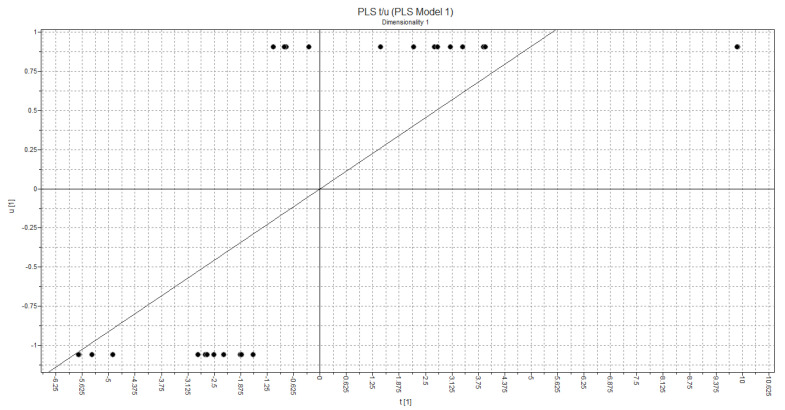
Arrangement of objects in relation to the activity of compounds.

**Figure 16 molecules-28-04009-f016:**
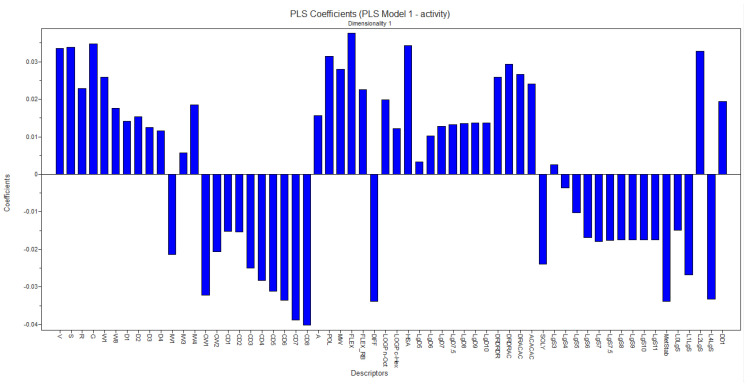
Coefficient graph generated from the PLS model.

**Table 1 molecules-28-04009-t001:** Other articles related to chalcones with anticancer activity.

Compounds	Number of Compounds under Study	Obtaining Method	In Vitro Activity	Cell Lineage	References
Pyrimidine derivatives linked to chloropyrazine	18	Synthesis	5 ± 1 µg/mL (IC50)	DU-145 (prostate cancer)	[75]
chalcone-thienopyrimidine	06	Synthesis	5.3 ± 2.1 µg/mL (IC50)	Hep-G2 (Hepatocarcinoma)	[76]
Natural chalcones	Various compounds	Literature review	-	Antimitotic activity (cell cycle inhibition in the G2/M phase)	[77]
1,3-diaryl-2-propen-1-one synthetic chalcones	Various compounds	Literature review	-	Inhibition of erythroid nuclear factor-related factor 2 (Nrf2)	[78]
Natural, synthetic, and semi-synthetic chalcones	Various compounds	Literature review	-	Biological activity of chalcones in negative undulating breast cancers (TNBCs)	[79]
Isoquinoline chalcone	01	Synthesis	Qualitative evaluation of marker expression	Evaluation of antioxidant and antiproliferative activity, as well as the action on p-53 and BAX markers	[80]
Spirooxindole hybrids	37	Synthesis	7 ± 0.27 µM (IC50) 5.5 ± 0.2 µM (IC50)	HCT-116 and HepG2 (colon cancer)	[81]
Pyrimidodiazepine derivatives containing the 2-chloro-4-anilinoquinazoline fragment	14	Synthesis	0.622 μM (GI50)	K-562 (Leukemia)	[82]
Benzothiazepine derivatives	20	Synthesis	16 ± 18 μg/mL (IC50) 12 ± 13 μg/mL (IC50) 15 ± 18 μg/mL (IC50)	HT-29 (colon cancer) MCF-7 (breast cancer) DU-145 (prostate cancer)	[83]
Homocyclic and heterocyclic chalcones	Various compounds	Literature review	-	Biological activity of chalcones in breast cancer.	[84]
Natural and synthetic chalcone derivatives	Various compounds	Literature review	50 ± 6 nM (GI50) 1–53.4 nM (GI50)	MDR A549/T (lung cancer) HCT-116/L (colon cancer) HL60/DOX (leukemia)	[85]
Natural and synthetic chalcone derivatives	Various compounds	Literature review	6.20 ± 2.82 μg/mL (IC50)	A-549 (lung cancer) HepG2 (colon cancer) MCF-7 (breast cancer) MDA-MB-231 (breast cancer) ALL-SIL (leukemia) SW1990 (pancreatic cancer) Vascular endothelial growth factor (VEGF) (anti-angiogenesis)	[86]
Derivatives of indole chalcones	Various compounds	Literature review	4 μM (IC50)	PaCa2 (pancreatic carcinoma) RT112 (bladder carcinoma)	[87]
Substituted natural chalcones	Various compounds	Literature review	-	Evaluation of the induction of apoptosis by the caspase-3 pathway	[88]
Natural chalcones present in the Sophora kingdom	Various compounds	Literature review	-	It does not mention a specific cell or mechanism; it only informs that the compounds have anticancer activity	[89]
Glycosidic derivatives of chalcones	Various compounds	Literature review	2.97 μM (IC50)	HL-60 (leukemia)	[90]
Natural and synthetic chalcone derivatives	Various compounds	Literature review	0.17–0.19 µM (IC50)	ABCG2 transport protein inhibition	[91]
Natural and synthetic chalcone derivatives	Various compounds	Literature review	-	It does not mention a specific cell or mechanism; it only informs that the compounds have anticancer activity.	[92]
Triazole-chalcone-conjugates	7	Synthesis	0.94–1.92 µM (IC50)	MCF-7 (breast cancer) Leukemia SR	[93]
Amino-naftil-chalcona	1	Synthesis	8 µg/mL (IC50)	U2OS (humanosteosarcoma cell line)	[94]
Polycyclic chalcone based acrylamides	4	Synthesis	38.46–48.25 µg/mL (IC50) 38.02–36.35 µg/mL (IC50)	MCF-7 (breast cancer) HeLa (cervical cancer)	[95]
Arylpropenone aminochalcone conjugates	17	Synthesis	6.7–9.8 µM (IC50)	MCF-7 (breast cancer)	[96]
Synthetic chalcones	Various compounds	Review article	-	NRF2, apoptosis, and BCL2	[97]
Natural and synthetic chalcone derivatives	Various compounds	Review article	-	Evaluation of the activity of chalcones in multiple mechanisms	[20]
Natural and synthetic chalcone derivatives	Various compounds	Review article	Survey of several studies and explanation of the mechanisms	Mechanisms related to gastric cancer	[98]
Synthetic chalcones	Various compounds	Review article about synthesis	Survey of several studies and explanation of the mechanisms	Various mechanisms	[19]
Natural and synthetic chalcone derivatives	Various compounds	Review articles on mechanisms of action of chalcones	Survey of several studies and explanation of the mechanisms	Various mechanisms	[22]
Natural and synthetic chalcone derivatives	Various compounds	Review articles on mechanisms the enzyme p53	Survey of several studies and explanation of the mechanisms	Action on p-53 protein	[99]
Chalcone hybrids	Various compounds	Review article on obtaining hybrid chalcone compounds combined with structure-activity studies	Survey of several studies and explanation of the mechanisms	The article emphasizes various synthetic compounds and various mechanisms to achieve cancer	[100]
Ferrocenyl chalcones	Various synthetic compounds	Review about application	Survey of several studies and explanation of the mechanisms	The article emphasizes various synthetic compounds and various mechanisms to achieve cancer	[101]
Natural chalcones	Various compounds	Review on compounds reported in specific mechanism	Compost survey with action on histones	Histone deacetylase	[31]
Natural flavans and (iso)flavanones	Various compounds	Review and application of synthetic compounds anticancer	Survey of several studies and explanation of the mechanisms	Various mechanisms	[102]
Chalcone based metal cordination	Various compounds	Review and application of synthetic compounds anticancer	Survey of several studies and explanation of the mechanisms	Various mechanisms for obtaining synthetic chalcones	[103]
Chalcone heterocycles synthesis	Various compounds	Review and application of synthetic chalcones in cancer	Various mechanisms for obtaining synthetic chalcones	Various mechanisms for obtaining synthetic chalcones	[25]
Quinoline chalcone hybrids	Various compounds	Review and application of synthetic chalcones in cancer	Various mechanisms for obtaining synthetic chalcones	Various mechanisms for obtaining synthetic chalcones	[104]
Flavonoids overview addressing natural chalcones	Various compounds	Review about application of phytochemical constituents	-	Blade cancer	[105]
Chalcones and other compounds	Various compounds	Review on bioisosterism and obtaining drugs	-	Various mechanisms	[106]
Coumarin-chalcone hybrids	Various compounds	Review and application of synthetic chalcones in cancer	Various mechanisms for obtaining synthetic chalcones	Various mechanisms	[107]
Phytochemical constituents of the Didymorcarpus wall (Gesneriaceae)	Various compounds	Review about application of phytochemical constituents	Survey of several studies and explanation of the mechanisms	Various mechanisms	[108]
Phytochemical constituents of the Kawa (piper methistycum)	Various compounds	Review about application of phytochemical constituents	Survey of several studies and explanation of the mechanisms	Various mechanisms	[109]
Flavonoids overview addressing natural chalcones	Various compounds	Review about application of phytochemical constituents	Survey of several studies and explanation of the mechanisms	Various mechanisms	[110]
Flavonoids overview addressing natural chalcones	Various compounds	Review article that addresses the antiviral activity for Herpes virus and a correlation with cancer pictures	Survey of several studies and explanation of the mechanisms	Antiviral activity and antitumor activity	[111]
Phenolic compounds of the *Mous alba*–natural chalcones	Various compounds	Review about application of phytochemical constituents	Survey of several studies and explanation of the mechanisms	Various mechanisms	[112]
Licochalcones	Various compounds	Review about application of phytochemical constituents	Survey of several studies and explanation of the mechanisms	Various mechanisms	[113]
Chalcone heterocycles synthesis	Various compounds	Review articles on mechanisms of action of chalcones	Survey of several studies and explanation of the mechanisms	Various mechanisms	[114]
Natural chalcones present in licorice–Chinese materia medica	Various compounds	Review articles on mechanisms of action of chalcones and other compounds	Survey of several studies and explanation of the mechanisms	Various mechanisms	[115]

## Data Availability

Not applicable.

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
