# Peer review of "Anticancer Activity of Chalcones and Its Derivatives: Review and In Silico Studies"

_molecules, 2023, doi:10.3390/molecules28104009_

Round 1
Reviewer 1 Report
The manuscript “Chalcones, dimers and derivatives with anticancer activity: a review” is not a review with deep processing of the material, but simply a data set. The authors do not explain why did they choose 2019-2023 years, what reviews were made before and etc. For example, a review 10.3390/biom11060894 is more detailed and devoted to the same topic, but the authors of the present review just ignore it. In my opinion, this review needs a serious revision and explanation of its importance and the place among other similar reviews. In the present form, I can just to recommend to reject this review.
Author Response
Dear Reviewer,
Thank you for your valuable feedback on our manuscript. We appreciate the time and effort you have taken to review our work. We understand the concerns you have raised and have addressed them as follows:
Reviewer 01:
Question |
The manuscript “Chalcones, dimers and derivatives with anticancer activity: a review” is not a review with deep processing of the material, but simply a data set. The authors do not explain why did they choose 2019-2023 years, what reviews were made before and etc. For example, a review 10.3390/biom11060894 is more detailed and devoted to the same topic, but the authors of the present review just ignore it. In my opinion, this review needs a serious revision and explanation of its importance and the place among other similar reviews. In the present form, I can just to recommend to reject this review.
|
Answer |
All review articles corresponding to the topic under study in the chosen time period were added. Initially, a description of the reviews that have already been published was performed in Table 1 and all articles referring to literature reviews were added. |
Reviewer 2 Report
1. Figure 03- the carbon atoms in the formula must be numbered.
2. Line 136 - I didn't find compound 13a in the figure 04.
3. Chemical compounds should be renumbered. The compound numbers cannot be the same.
4. Lines 154-157: “who reported the synthesis of chalcones via aldol condensation between bis(2-(2-methoxyphenyl)thiazolidin-4-one) with p-methoxybenzaldehydes catalyzed by NaOAc. The compounds were cyclized with thiazoles under reflux in AcOH 156 generating functionalized chalcone derivatives.”- wrong description of the synthesis, see original paper.
5. Line 159 - no IC50 unit
6. Figure 05- the compound (1) is not a chalcone.
7. Line 286- it should be ethoxychalcones
8. Many errors in the text. It should be thoroughly checked.
9. Line 426-427: “O composto (2) (Figure 13) apresentou os menores valores de IC50 (2,1 e 2,4 μg.mL-1 426 contra SF-295 e PC-3, respectivamente).”- ???
10. Figure 14- lack of an oxygen atom in the structure.
11. Chemometric analyses- I am not able evaluate this chapter.
12. The way of notation of units, for example ml and mL, should be unified.
13. Citations in the literature chapter should be standardized.
14. The arrangement of individual compounds in the figures needs to be refined.
15. Line 378: “2-((E)-4-((E)-3-oxo-3-(p-tolyl)prop-1-en-1-yl)benzylidene)hydrazine-1-carboximidamide” - is this long name really necessary ?
Overall, I find the publication interesting.
Author Response
Dear Reviewer,
Thank you for your valuable feedback on our manuscript. We appreciate the time and effort you have taken to review our work. We understand the concerns you have raised and have addressed them as follows:
Question 1. |
Figure 03- the carbon atoms in the formula must be numbered. |
Added |
|
Question 2. |
Line 136 - I didn't find compound 13a in the figure 04. |
Removed |
|
Question 3. |
Chemical compounds should be renumbered. The compound numbers cannot be the same. |
Changed |
|
Question 4. |
Lines 154-157: “who reported the synthesis of chalcones via aldol condensation between bis(2-(2-methoxyphenyl)thiazolidin-4-one) with p-methoxybenzaldehydes catalyzed by NaOAc. The compounds were cyclized with thiazoles under reflux in AcOH 156 generating functionalized chalcone derivatives.”- wrong description of the synthesis, see original paper. |
Removed |
|
Question 5. |
Line 159 - no IC50 unit |
Changed |
|
Question 6. |
Figure 05- the compound (1) is not a chalcone. |
The compound in Figure 5 was removed from the main text of the manuscript for Table 1, as it is a synthetic derivative of chalcone, which has a different structure, which, according to the authors Srinivas and Rajitha (2022), the synthesis of chalcones with subsequent cyclization of the α,β-unsaturated system, resulting in heterocyclic derivatives.
Reference: Srinivas, A.; Rajitha, S.R. Synthesis of New Heterocycles via Methylenebis (2-(2-Methoxyphenyl) Thiazolidin-4-One) as Potential Anticancer Agents. Org. Commun. 2022, 15. |
|
Question 7. |
Line 286- it should be ethoxychalcones |
Adjusted |
|
Question 8. |
Many errors in the text. It should be thoroughly checked. |
Adjusted |
|
Question 9. |
Line 426-427: “Compound (2) (Figure 13) had the lowest IC50 values (2.1 and 2.4 μg.mL-1 426 against SF-295 and PC-3, respectively).”- ? ?? |
Adjusted |
|
Question 10. |
Figure 14- lack of an oxygen atom in the structure. |
Adjusted |
|
Question 11. |
The way of notation of units, for example ml and mL, should be unified. |
Changed |
|
Question 12. |
Citations in the literature chapter should be standardized. |
Changed |
|
Question 13. |
The arrangement of individual compounds in the figures needs to be refined. |
Changed |
|
Question 14. |
Line 378: “2-((E)-4-((E)-3-oxo-3-(p-tolyl)prop-1-en-1-yl)benzylidene)hydrazine-1-carboximidamide” - is this long name really necessary ? |
Changed |
Reviewer 3 Report
The manuscript by Ferreira Leite, F. et al. described a bibliographic review on the anticancer activity of chalcones in vitro in the period between the years 2019 to 2023. The overall flow and structure of the text are well written. Although this work has been competently carried out, a point to be considered is the numbering of the structures consecutively, never mind the Figure numbering. Also, the resolution of the structures and their respective labels.
Despite that, the following issues should be addressed in a (major) revision:
1) It is indicated that 159 articles were found, but there are only 101 references. Explain why the remaining ones were not considered.
2) Typos, delete in biological abstract line 22.
3) Page 1., line 38 use other references such as: https://doi.org/10.1002/asia.202200706. https://doi.org/10.1080/17460441.2019.1573812
10.2174/1871520620999201124212840
https://doi.org/10.3390/biom11060894
https://doi.org/10.1080/14786419.2021.2000980
In some cases, the difference between this work and what is published should be indicated.
4) Figure 1, delete 01, consider this throughout the document, on the other hand, place reference and remove source (see author guidelines).
5) Page 14. Line 136, check numbering of compound 14a.
6) Pag. 5, line 158, take care of the numbering, the number 1 is repeated in other Figures, give consecutive numbering.
7) Figure 5, this structure is not a derived chalcone, if so, justify from the structural point of view.
8) Page 5 Figure 1 (triazole) should be verified, a double bond is missing.
9) Figure 6. Place superscripts on the R and verify names (triflouoro) and color numbering for easy identification.
10) Page 7. Line 210, Hybrids, in lower case.
11) The word In vitro should be homogenized in some are in italics and others are not (abstract, keywords, Pag, 8, line 270).
12) Pag. 9, line 297 NO2, superscript. Pag. 10, line 338 and 339
13) Fig. 10, improve image resolution.
14) Pag. 17, line 456, it indicates compounds but does not indicate which Figure. Please take care of numbering.
15) Page 17, line 461, delete point.
16) Verify English writing, for example: pag. 13, line 426
17) In Table 1, it is recommended to change names (compounds) to a generic structure.
Author Response
Dear Reviewer,
Thank you for your valuable feedback on our manuscript. We appreciate the time and effort you have taken to review our work. We understand the concerns you have raised and have addressed them as follows:
Question 01. |
It is indicated that 159 articles were found, but there are only 101 references. Explain why the remaining ones were not considered. |
Exclusion criteria were articles unrelated to chalcones, articles unrelated to anticancer activity and articles with unavailable access. |
|
Question 02. |
Typos, delete in biological abstract line 22. |
Adjusted |
|
Question 03. |
Page 1., line 38 use other references such as: https://doi.org/10.1002/asia.202200706. https://doi.org/10.1080/17460441.2019.1573812 10.2174/1871520620999201124212840 https://doi.org/10.3390/biom11060894 https://doi.org/10.1080/14786419.2021.2000980 In some cases, the difference between this work and what is published should be indicated. |
Added |
|
Question 04. |
Figure 1, delete 01, consider this throughout the document, on the other hand, place reference and remove source (see author guidelines). |
Changed |
|
Question 05. |
Page 14. Line 136, check numbering of compound 14a. |
Added |
|
Question 06. |
Pag. 5, line 158, take care of the numbering, the number 1 is repeated in other Figures, give consecutive numbering. |
Changed |
|
Question 07. |
Figure 5, this structure is not a derived chalcone, if so, justify from the structural point of view. |
The compound in Figure 5 was removed from the main text of the manuscript for Table 1, as it is a synthetic derivative of chalcone, which has a different structure, which, according to the authors Srinivas and Rajitha (2022), the synthesis of chalcones with subsequent cyclization of the α,β-unsaturated system, resulting in heterocyclic derivatives.
Reference: Srinivas, A.; Rajitha, S.R. Synthesis of New Heterocycles via Methylenebis (2-(2-Methoxyphenyl) Thiazolidin-4-One) as Potential Anticancer Agents. Org. Commun. 2022, 15. |
|
Question 08. |
Page 5 Figure 1 (triazole) should be verified, a double bond is missing. |
Adjusted |
|
Question 09. |
Figure 6. Place superscripts on the R and verify names (triflouoro) and color numbering for easy identification. |
Adjusted |
|
Question 10. |
Page 7. Line 210, Hybrids, in lower case. |
Adjusted |
|
Question 11. |
The word In vitro should be homogenized in some are in italics and others are not (abstract, keywords, Pag, 8, line 270). |
Adjusted |
|
Question 12. |
Pag. 9, line 297 NO2, superscript. Pag. 10, line 338 and 339 |
Changed |
|
Question 13. |
Fig. 10, improve image resolution. |
Improved |
|
Question 14. |
Pag. 17, line 456, it indicates compounds but does not indicate which Figure. Please take care of numbering. |
Changed |
|
Question 15. |
Page 17, line 461, delete point. |
Changed |
|
Question 16. |
Verify English writing, for example: pag. 13, line 426 |
Adjusted |
|
Question 17. |
In Table 1, it is recommended to change names (compounds) to a generic structure. |
Changed |
Round 2
Reviewer 1 Report
The authors made a good revision work on the manuscript, and now it looks acceptable.
Hiwever, I have some remarks:
1. Part "2. Overview of review articles on chalcones: Approaches and Perspectives" needs serious stylish and language check.
Author Response
We thank the reviewer for all his comments and suggestions that improved the manuscript during all process.
As request - part 2 was reviewed by a native english speaker
Reviewer 2 Report
It is OK.
Author Response
We thank the reviewer for all his comments and suggestions that improved the manuscript
Reviewer 3 Report
The authors have addressed most of the issues I commented in my previous report in a satisfactory manner.
However, there are still issues that are not convincing:
· Dimers in the title (and pag 3, line 100) should be reconsidered, in the text there is no special emphasis.
· Please, authors are kindly requested to carefully and thorough and careful analysis of the modifications suggested and made to avoid typos, for example: Figure 02, page 2, line 82 Figure 03, in vitro (some in italics and some not), Butein (pag. 5, line 201)
· Homogenize documents when using acronyms, some of them are indicated when they appear for the first time, others are not. It is recommended to check the author's guidelines or make a list of them. PLS (pag. 2 line 93), MDR, NOTCH, etc.
· Reconsider the title of item 2: “Overview of review articles on chalcones: Approaches and Perspectives” to Approaches of review.
· Verify typos: Pag. 2, line 80. Pag 2, line 82 Figure 03, eliminated 0, as well Figure 06 (pag 7, line 236, 254; pag 6, line 231); Pag.4 line 117, ati-inflammatory; Pag 4, lines 151, Di-els-Alder; Pag 5, line157, in vivo should be in vivo (pag 12, line 403). Pag 6, line 216, 14a was eliminated, but what was the new number?
· Pag 7, line 254, what´s means, CH1, CH2??? (pag 7, line 253 and 254).
· Homogenize decimals throughout the document (pag 11, line 355 (6.555–10.14 μM.)
· The numbering of each structure must be aligned and homogeneous for the entire document.
· Check, pag 13, line 417, “cy-totoxi-city against Raw-264.7”
· Verify all bibliography: Corsini and collaborators (2020) [62], it is described as: Kudličková, Z.; Takáč, P.; Sabolová, D.; Vilková, M.; Baláž, M.; Béres, T.; Mojžiš, J. Novel 1-Methoxyindole-and 2-Alkoxyindole-Based Chalcones: Design, Synthesis, Characterization, Antiproliferative Activity and DNA, BSA Binding Interactions. Med. Chem. Res. 2021, 30, 897–912. Therefore, all bibliography should be verified.
· Proper use of upper- and lower-case letters throughout the document (pag 13, line 431: Histone lysine demethylase; Chlorine, Bromine and Fluorine, pag 26, line 606).
· Pag 24, line 548 to 551, a series of compounds is described but no figure is indicated, or in which figures they are found, there is uncertainty when making the corresponding modifications, it is recommended to make a figure where they are combined.
· Table 01??
· Pag 24, line 556, the point should be deleted,” statistical index. in”
Author Response
We thank the reviewer for all his comments and suggestions that improved the manuscript during all process.
- Dimers in the title (and pag 3, line 100) should be reconsidered, in the text there is no special emphasis.
Changed title and modified item 2.
- Please, authors are kindly requested to carefully and thorough and careful analysis of the modifications suggested and made to avoid typos, for example: Figure 02, page 2, line 82 Figure 03, in vitro (some in italics and some not),Butein (pag. 5, line 201)
Figures and numbering have been corrected, the words “in vitro” have been standardized in italics, as well as “Butein” has been standardized.
- Homogenize documents when using acronyms, some of them are indicated when they appear for the first time, others are not. It is recommended to check the author's guidelines or make a list of them. PLS (pag. 2 line 93), MDR, NOTCH, etc.
Acronym descriptions were added and the text was homogenized.
- Reconsider the title of item 2: “Overview of review articles on chalcones: Approaches and Perspectives” to Approaches of review.
Thanks for the suggestion, we changed the title of item 2.
- Verify typos: Pag. 2, line 80. Pag 2, line 82 Figure 03, eliminated 0, as well Figure 06 (pag 7, line 236, 254; pag 6, line 231); Pag.4 line 117, ati-inflammatory; Pag 4, lines 151, Di-els-Alder; Pag 5, line157, in vivo should be in vivo(pag 12, line 403). Pag 6, line 216, 14a was eliminated, but what was the new number?
The words were standardized and figure 14a refers to compound 5 in figure 4 (adjusted).
- Pag 7, line 254, what´s means, CH1, CH2??? (pag 7, line 253 and 254).
A typo occurred, they refer to compounds 9 and 10 respectively. The error has been corrected.
- Homogenize decimals throughout the document (pag 11, line 355 (6.555–10.14 μM.)
Adjusted
- The numbering of each structure must be aligned and homogeneous for the entire document.
Adjusted
- Check, pag 13, line 417, “cy-totoxi-city against Raw-264.7
Adjusted
- Verify all bibliography: Corsini and collaborators (2020) [62], it is described as: Kudličková, Z.; Takáč, P.; Sabolová, D.; Vilková, M.; Baláž, M.; Béres, T.; Mojžiš, J. Novel 1-Methoxyindole-and 2-Alkoxyindole-Based Chalcones: Design, Synthesis, Characterization, Antiproliferative Activity and DNA, BSA Binding Interactions. Chem. Res. 2021, 30, 897–912. Therefore, all bibliography should be verified.
References have been checked and corrected.
- Proper use of upper- and lower-case letters throughout the document (pag 13, line 431: Histone lysine demethylase; Chlorine, Bromine and Fluorine, pag 26, line 606).
Adjusted
- Pag 24, line 548 to 551, a series of compounds is described but no figure is indicated, or in which figures they are found, there is uncertainty when making the corresponding modifications, it is recommended to make a figure where they are combined.
Figure 14 was added referring to the series of compounds under study. It is divided into two sections, the first (Section A) corresponding to the active compounds and the second (Section B) corresponding to the inactive compounds.
- Table 01??
The indication of the table number was corrected according to the journal's requirements.
- Pag 24, line 556, the point should be deleted,” statistical index. in”
The point has been deleted.